# PREMISE SELECTION FOR A LEAN HAMMER

**Thomas Zhu[1,*], Joshua Clune[1,*],**
**Jeremy Avigad[1,†], Albert Q. Jiang[2,†], Sean Welleck[1,†]**
[1]Carnegie Mellon University, [2]Mistral AI
[*]Equal contribution, [†]Equal advising
{thomaszh,jclune,avigad,swelleck}@andrew.cmu.edu, qj213@cam.ac.uk

## ABSTRACT

Neural methods are transforming automated reasoning for proof assistants, yet integrating these advances into practical verification workflows remains challenging. A *hammer* is a tool that integrates premise selection, translation to external automatic theorem provers, and proof reconstruction into one overarching tool to automate tedious reasoning steps. We present LEANPREMISE, a novel neural premise selection system, and we combine it with existing translation and proof reconstruction components to create LEANHAMMER, the first end-to-end domain general hammer for the Lean proof assistant. Unlike existing Lean premise selectors, LEANPREMISE is specifically trained for use with a hammer in dependent type theory. It also dynamically adapts to user-specific contexts, enabling it to effectively recommend premises from libraries outside LEANPREMISE's training data as well as lemmas defined by the user locally. With comprehensive evaluations, we show that LEANPREMISE enables LEANHAMMER to solve 21% more goals than existing premise selectors and generalizes well to diverse domains. Our work helps bridge the gap between neural retrieval and symbolic reasoning, making formal verification more accessible to researchers and practitioners.[1]

## 1 INTRODUCTION

Interactive proof assistants have long been used to verify the correctness of hardware, software, network protocols, cryptographic protocols, and other computational artifacts. Buoyed by successes like the Liquid Tensor Experiment (Lean Community, 2022) and the formalization of the Sphere Eversion Theorem (van Doorn et al., 2023), mathematicians are increasingly using the technology to verify mathematical theorems (Tao, 2023) and build substantial mathematical libraries (The Mathlib Community, 2020).

When working with a proof assistant, a user describes a proof in an idealized proof language, which is a programming language that provides sufficient detail for the computer to construct a precise formal derivation in the proof assistant's underlying axiomatic system. One of the challenges to formalization is the requirement to spell out what seem like straightforward inferences in painful detail. This problem is exacerbated by the fact that at the most basic level of interaction, users are required to name the required premises (i.e., definitions and lemmas) explicitly to justify an inference step, from a library of hundreds of thousands of previously derived facts.

A *hammer* (Meng et al., 2006; Paulson & Blanchette, 2012; Blanchette et al., 2016) is a tool designed to ease the pain of formalization by filling in small inferences automatically. Typically, a hammer has three components: given a goal to prove, one first selects a moderate number of premises from the library, project files, current file, and hypotheses that, one hopes, are sufficient to prove the goal. This is known as *premise selection*. Then one translates the premises and the goal into the language of powerful external automated theorem provers like Vampire (Kovács & Voronkov, 2013), E (Schulz et al., 2019), and Zipperposition (Vukmirović et al., 2022), or SMT solvers like Z3 (de Moura & Bjørner, 2008) and cvc5 (Barbosa et al., 2022). Finally, if the external prover succeeds in proving

---

[1]LEANPREMISE is available at https://github.com/hanwenzhu/premise-selection and LEAN-HAMMER is available at https://github.com/JOSHCLUNE/LeanHammer.

the goal, it reports back the specific premises used, from which a formal proof in the proof assistant is reconstructed.

In this paper, we present LEANPREMISE, a new premise selection tool for the Lean proof assistant (de Moura & Ullrich, 2021). We combine it with the DTT-to-HOL (dependent type theory to higher-order logic) translation tool, Lean-auto (Qian et al., 2025), and internal proof-producing tactics, Duper (Clune et al., 2024) and Aesop (Limperg & From, 2023), resulting in LEANHAMMER, the first end-to-end domain general hammer for Lean. Through comprehensive evaluations, we show that LEANHAMMER can hit nails.

Our work, which extends methods of premise selection used by Magnushammer (Mikuła et al., 2024) and LeanDojo (Yang et al., 2023), is therefore an auspicious combination of neural premise selection methods with symbolic proof search. For the first time, we specifically design contrastive learning methods for the first end-to-end domain general hammer in Lean. We explain the design choices to make LEANPREMISE performant for LEANHAMMER, including new *hammer-aware data extraction* techniques. An important feature of LEANPREMISE is that it dynamically augments the library of facts with locally defined facts from the user's project, which is essential in practice.

Our core contributions are as follows:

- We develop LEANPREMISE, a premise selection tool for a hammer in dependent type theory.
- We combine LEANPREMISE with Aesop, Lean-auto, and Duper to make LEANHAMMER, the first domain general hammer in Lean.
- We provide an accessible user-facing tactic interface that can dynamically process new premises in the environment.
- We conduct comprehensive evaluations of LEANHAMMER's performance on Mathlib and its ability to generalize to miniCTX-v2 (Hu et al., 2025). Through these evaluations, we show that LEAN-HAMMER solves 21% more goals with LEANPREMISE than with existing premise selectors and that LEANPREMISE enables LEANHAMMER to effectively use libraries and premises it hasn't seen before.

Note that premise selection can be used in other ways, for example, for calling various types of internal automation directly, for presenting suggestions to a user engaged in manual proof, or for use in a neural or neurosymbolic search. Although our focus here has been on a hammer, we expect that many of the methods we develop carry over to other settings.

## 2 RELATED WORK

### 2.1 HAMMERS IN INTERACTIVE PROOF ASSISTANTS

As explained in the introduction, hammers support interactive proving by completing small inferences, called *goals*. The first and still most successful hammer in use today is Isabelle's Sledgehammer, developed initially by Meng et al. (2006) and further developed by Paulson & Blanchette (2012); Blanchette et al. (2013), and many others. Since then hammers have been developed for HOL (Kaliszyk & Urban, 2015a), Mizar (Kaliszyk & Urban, 2015b), Rocq (Czajka & Kaliszyk, 2018), and Metamath (Carneiro et al., 2023), among others. Of these, only Rocq is based on dependent type theory. Despite Lean's popularity, no hammer has been developed for Lean.

### 2.2 NEURAL THEOREM PROVING

Numerous neural-network-based tools have been developed to prove theorems. A straightforward approach of using neural models is to let them generate steps in proofs, notable examples of which include GPT-*f* (Polu et al., 2023), HTPS (Lample et al., 2022), ReProver (Yang et al., 2023), DeepSeek-Prover (Xin et al., 2024a;b) for Lean, LISA (Jiang et al., 2021) and Thor (Jiang et al., 2022) for Isabelle, and PALM (Lu et al., 2024), Cobblestone (Kasibatla et al., 2024), and Graph2Tac (Blaauwbroek et al., 2024) for Coq/Rocq. Another line of work uses neural models to generate entire proofs or proof sketches (Jiang et al., 2023; Zhao et al., 2023; Wang et al., 2024; First et al., 2023; Lin et al., 2025a;b; Wang et al., 2025; Chen et al., 2025). These proof search approaches are complementary to a hammer, which serves as a tactic that may be used by neural models.

Hammers are embedded in a number of neural theorem proving frameworks such as Thor and Draft, Sketch, and Prove (Jiang et al., 2022; Zhao et al., 2023; Jiang et al., 2023; Wang et al., 2024) to fill small gaps in the proofs. It is worth noticing that all these works use the Isabelle proof assistant (Nipkow et al., 2002) where the communication infrastructure (Jiang et al., 2021) between neural models and the proof assistant is relatively mature and hammering is easy to set up. Our work makes calling a hammer in Lean possible.

Despite a large number of research works, practical tools that a working mathematician has access to without complex setup or prohibitive costs remain scarce. Recent state-of-the-art methods use reinforcement learning on e.g. 7B LLMs with thousands of passes for a single theorem and use infrastructures not callable from Lean (Wu et al., 2024; Lin et al., 2025a;b; Dong & Ma, 2025; Chen et al., 2025), so it is prohibitive for Lean users to train, test, or use them. Our work brings forward a tool that is packaged as a tactic and can be called straightforwardly from any IDE for Lean with low computational cost and latency, hence enabling better automation for the masses.

## 2.3 PREMISE SELECTION

Formalizing mathematics in proof assistants requires users to select relevant premises from libraries of hundreds of thousands of facts. To help facilitate this task, premise selection has been developed for a variety of proof assistants, using both neural and symbolic techniques. MePo (Meng & Paulson, 2009) is a symbolic premise selector which has been widely used in Isabelle's Sledgehammer. Other premise selectors which target hammers but use traditional machine learning techniques include MaSh (Kühlwein et al., 2013), $k$-NN based premise selection for HOL4 (Gauthier & Kaliszyk, 2015), CoqHammer's premise selection (Czajka & Kaliszyk, 2018), and random forest based premise selection for Lean (Piotrowski et al., 2023). LEANPREMISE differs from these by using modern LM-based retrieval methods.

(L)LM-based premise selection trained by contrastive learning has also been explored for a variety of use cases. Lean State Search (Tao et al., 2025) recommends relevant premises directly to Lean users. Magnushammer (Mikuła et al., 2024) generates premises to supply directly to proof reconstruction tactics. ReProver and Lean Copilot (Yang et al., 2023; Song et al., 2024) retrieve premises to augment neural next-tactic generation. Unlike these, LEANPREMISE is specifically designed with hammer integration in mind, which requires specific data extraction and loss formulation, and the resulting selector to be fast and domain general.

## 3 METHODS

### 3.1 LEANHAMMER PIPELINE

Hammers broadly consist of three primary components: premise selection, translation to external automatic theorem provers, and proof reconstruction. In traditional hammer pipelines, such as Isabelle's Sledgehammer, these components are composed in a linear fashion, with the premises from premise selection informing the translation to automatic theorem provers and the output from automatic theorem provers informing proof reconstruction. In other works, such as Magnushammer (Mikuła et al., 2024) and Lean Copilot (Song et al., 2024), premises from premise selection are provided directly to proof reconstruction tactics or language models, without translating and sending them to external automatic theorem provers. LEANHAMMER introduces a new, unified hammer pipeline that uses premise selection in both of these ways.

Figure 1 gives an overview of the LEANHAMMER pipeline. In addition to LEANPREMISE itself, LEANHAMMER is built upon Aesop, Lean-auto, and Duper. Aesop is a highly extensible proof search tool that can be augmented with new proof search rules and procedures. Lean-auto is a translation tool that does not search for proofs itself, but instead translates dependently typed Lean goals into higher-order logic problems which can be solved by external automatic theorem provers such as Zipperposition. Finally, Duper is a less powerful but proof-producing proof search tool which implements many of the techniques found in automatic theorem provers, and is therefore well suited to rediscovering and verifying proofs found by external automatic theorem provers.

In broad strokes, Aesop is called first and prioritizes finding a proof using its own built-in rules. If a short proof using only built-in rules is not found quickly, it explores direct premise applications

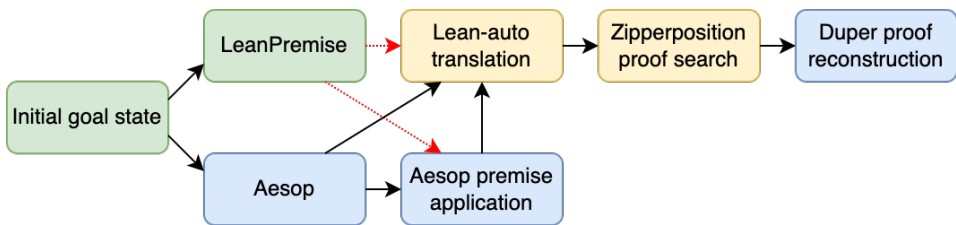

Figure 1: Overview of the LEANHAMMER pipeline. Phases that can neither fail nor produce a terminal proof are green, phases that can fail but cannot produce a terminal proof are yellow, and phases that can produce a terminal proof are blue. Black solid arrows indicate control flow, while red dashed arrows indicate the transfer of information between phases.

using premises recommended by LEANPREMISE,[2] and it queries Lean-auto to see if subgoals can be closed using premises from the selector.[3] When Lean-auto is given a subgoal, it translates that subgoal (along with the premises provided by the selector) to higher-order logic and interfaces with Zipperposition to find a proof. If Zipperposition succeeds, then Duper is provided just the set of premises used by Zipperposition to solve the translated problem, and Duper attempts to reconstruct a proof from these premises. For an illustrative example of LEANHAMMER's pipeline in action, see the extended version of this paper (Zhu et al., 2025, Section B).

## 3.2 DATA EXTRACTION

To support LEANPREMISE, we develop a data extraction pipeline designed to gather not just information useful for next-tactic generation or human examination, but all of the information that may be helpful for a hammer tasked with discovering an end-to-end proof. This pipeline is used dynamically to extract premises that LEANPREMISE can retrieve at runtime, including premises or definitions defined by the user locally, and it is used statically to extract (state, premise) pairs for training. As we describe our data extraction pipeline, we note the measures taken to collect data that go beyond what appears explicitly in the source code for the formal proofs.

### 3.2.1 SIGNATURE EXTRACTION

A key aspect of premise selectors is how premises are presented to the model. Previous work (Yang et al., 2023) extracts raw strings from the source code, which ignores many details in the full signature (see Section A of the extended version of this paper (Zhu et al., 2025) for an example). We adopt a new *normalized serialization* as follows. For each theorem and definition in each module, we extract the documentation description in the source code (its docstring), if it exists, as well as its kind (theorem or definition), name, arguments, and overall type. Together, these can be composed into a signature of the form `docstring? kind name arguments* : type`. When converting these signatures into strings, we disable notation pretty printing (e.g. we print $\mathbb{N}$ as `Nat`), and we print every constant with its fully qualified name (e.g. we print `I` as `Complex.I`). This standardizes premise representation, so that it depends only on the type of the premise and does not depend on open namespaces, custom notations, and surface-level syntax, which may change at run time. For an illustrative example, see Section A of the extended version of this paper (Zhu et al., 2025).

The signatures extracted in this manner are used to form the set of premises $\mathcal{P}$ that LEANPREMISE is allowed to retrieve from. This signature extraction pipeline is also used to dynamically extract new premises at runtime (Section 3.3.2). To prevent LEANPREMISE from constantly recommending theorems that are technically relevant to the goal but never useful for our hammer's automation, we filter out a blacklist of 479 basic logic theorems such as `and_true` from $\mathcal{P}$. We also filter out Lean language-related (e.g. metaprogramming) definitions not useful for proofs.

---

[2]Premise applications are rules added to Aesop of the form (add unsafe 20% <premise>) where <premise> is a premise selected by the premise selector.

[3]Lean-auto is added to Aesop as a rule of the form (add unsafe 10% (by auto [*, <premises>])) where <premises> is a list of premises selected by the premise selector.

### 3.2.2 STATE AND PREMISE EXTRACTION

The next key question is which (state, premise) pairs are extracted from human-written proofs to train the model. Previous premise selectors (Yang et al., 2023) that focus on tactic generation only extract from tactic-style proofs, and only extract explicit premises appearing in the raw source code of only the next tactic. Our *hammer-aware data extraction* improves upon this in several ways. First, we extract from both term-style and tactic-style proofs, significantly increasing training samples especially for short proofs that LEANHAMMER is intended to automate. Second, for multi-tactic proofs, the model is trained to select premises to close the goal (all tactics) rather than to modify the goal (first tactic), because hammers are designed to finish proofs. Third, we extract both implicit and explicit premises from the proof, including ones implicitly called by automation such as `simp`. Finally, we format states with the same normalized serialization as for premises.

Specifically, for each theorem in each module, we collect data on the premises used to prove it. Additionally, for each theorem proven via tactic-style proofs, we collect data on all intermediate goal states induced by the tactic sequence. For an illustrative example, see Section A of the extended version of this paper (Zhu et al., 2025). Ultimately, all data we extract contains:

- A proof state obtained either from the beginning of a theorem or from an intermediate step of a tactic-style proof.
- The name and signature of the theorem from which the state was extracted.
- The set of premises used to prove the theorem[4].

When theorems are proven via term-style proofs, meaning the theorem's proof term is explicitly written in the source code, the set of premises we extract is the set of theorems that appear in the proof term. When theorems are proven via tactic-style proofs, meaning automation is invoked to tell Lean how to build a proof term, the set of premises we extract contains both the theorems that appear in the proof term constructed by the tactic sequence (so that all implicit premises are collected), as well as any theorems and definitions that are explicitly used in `rw` or `simp` calls.

The benefit of collecting explicit theorems and definitions from `rw` and `simp` calls relates to Lean's dependent type theory. In Lean, terms can be definitionally equal without being syntactically equal, and because of this, tactic-style proofs can invoke definitional equalities that do not appear in final proof terms. We therefore collect these definitional equality premises. We experimentally verify that our hammer-aware data extraction benefits LEANHAMMER in Section 4.4.

### 3.3 PREMISE SELECTION

LEANPREMISE uses the standard method of retrieval using textual encoders. In order to retrieve $k$ premises for a state $s$, we first determine the set $\mathcal{P}_s$ of accessible premises at position $s$, comprising lemmas and definitions that are imported from other modules or declared earlier in the file. We use an encoder-only transformer model $E$ to embed both the state $s$ and every premise $p \in \mathcal{P}_s$, and the resulting set of premises retrieved is

$$\texttt{select\_premises}(s, k, \mathcal{P}_s) = \text{top-}k_{p \in \mathcal{P}_s} \texttt{sim}(E(s), E(p)) \tag{1}$$

where $\texttt{sim}(u, v) = u^\top v / \|u\|_2 \|v\|_2$ is cosine similarity. In Section 3.3.2 we describe the mechanism for caching embedding and quick retrieval of the premises.

We do not train a separate reranking model as in Mikuła et al. (2024), because we did not find it to increase performance in early experiments, especially since a hammer favors recall much more than precision, and we determined the optimal $k$ to be at least 16, at which point reranking does not offer much improvement. It is also costly to deploy in practice.

### 3.3.1 MODEL TRAINING

We use a modified version of the InfoNCE loss (Oord et al., 2018) to train the encoder model. On a high level, each batch consists of (state, premise) pairs, and a contrastive loss is used to let the

---

[4]We also experimented with pairing states with just the set of premises used to close said states, as opposed to all premises used to prove the overall theorem, but our preliminary experiments showed that this yielded worse results than including all premises.

| Premise selector | LM-based | Callable in Lean | New premises |
|---|:---:|:---:|:---:|
| ReProver (Yang et al., 2023) | ✓ | ✗ | ✗ |
| Lean Copilot (Song et al., 2024) | ✓ | ✓ | ✗ |
| Random forest (Piotrowski et al., 2023) | ✗ | ✓ | ✗ |
| MePo (Meng & Paulson, 2009) | ✗ | ✓ | ✓ |
| **LEANPREMISE** | ✓ | ✓ | ✓ |

Table 1: Usability comparison of existing premise selection tools. Note that this is orthogonal to the quantitative performance comparisons (Table 2).

model learn to select the correct premise out of all premises in this batch. One problem is that there are many premises in the library that do not appear in any proof. This is mitigated by also sampling negative premises in each batch (Mikuła et al., 2024; Yang et al., 2023). Another problem is that there are many premises that are shared across many proofs, so not all premises in the batch are negative. We use the following masked contrastive loss to address these problems.

Specifically, for each training step, we sample a batch of $B$ (state, premise) pairs, each consisting of a state $s_i$ and a premise $p_i^+ \in \mathcal{P}_{s_i}^+$ where $\mathcal{P}_{s_i}^+$ is the set of ground-truth premises for $s_i$ extracted as in Section 3.2. For each such pair $(s_i, p_i^+)$, we sample $B^-$ negative premises $\{p_{ij}^-\}_{j=1}^{B^-} \subseteq \mathcal{P}_{s_i} \setminus \mathcal{P}_{s_i}^+$, giving $B$ states and $B(1 + B^-)$ premises in total in each batch. Of these premises, we determine the set $\mathcal{N}_i = \{p_i^+\}_i \cup \{p_{ij}^-\}_{ij} \setminus \mathcal{P}_{s_i}^+$ of negative premises for state $s_i$, and mask out the positive ones in the loss to avoid mislabeling. The loss is:

$$\mathcal{L}(E) = \frac{1}{B} \sum_{i=1}^{B} \frac{\exp(\mathtt{sim}(E(s_i), E(p_i^+))/\tau)}{\exp(\mathtt{sim}(E(s_i), E(p_i^+))/\tau) + \sum_{p_i^- \in \mathcal{N}_i} \exp(\mathtt{sim}(E(s_i), E(p_i^-))/\tau)} \quad (2)$$

where $\tau$ is a scalar temperature hyper-parameter (set to 0.05 in our experiments).

### 3.3.2 API INTEGRATION

In order to make LEANPREMISE and LEANHAMMER more accessible for Lean users as well as downstream methods, we design our pipeline to maximize usability—it is directly callable in Lean, able to take in new premises, and efficient to run. Our pipeline for premise selection is as follows: when a user invokes premise selection, the client side (Lean) collects all currently defined premises $\mathcal{P}_s$ defined in the environment and the current proof state $s$ and sends them to a server that hosts the embedding model. The server embeds both the proof state and the list of premises, and then runs FAISS (Douze et al., 2024) on the premises to compute $\mathtt{select\_premises}(s, k, \mathcal{P}_s)$, and returns this list of $k$ premises back to the client. Since the typical size of $\mathcal{P}_s$ is on the scale of $\sim$70k, the server also caches the embeddings of premises at fixed versions of Mathlib, and only recomputes embeddings of signatures of new premises uploaded by the user (e.g. when working outside Mathlib or when the user has new premises in the context); the client side also caches the signatures of these new premises computed as in Section 3.2.1.

LEANHAMMER is built as a tactic that can be directly called in Lean. It calls LEANPREMISE as a subprocedure and the retrieved premises are then input to the LEANHAMMER pipeline (Section 3.1). In Mathlib, premise selection usually takes about 1 second amortized on a CPU server (and well under 1 second for a single-GPU server). The full LEANHAMMER pipeline on average takes well under 10 seconds (see Section 4.2).

To the best of our knowledge, LEANPREMISE is the first premise selector using language models that can be directly invoked in Lean and can incorporate new user-defined premises. It is also efficient to run and requires no system setup for the user, because the main computation is only a few string embeddings, and done centrally in a server by default. This makes the premise selector itself a desirable user-facing tactic for the Lean community. Similarly, the full LEANHAMMER can be called straightforwardly in Lean as a tactic. This bridges a gap that many previous LM-based retrievers and provers leave. See Table 1 for a comparison.

### 3.4 VARIATIONS AND EXTENSIONS

Here, we discuss variations on LEANHAMMER's design, implemented as settings that can be controlled by the user. Note that the pipeline described in Section 3.1 has premises input both to Aesop as premise application rules and to Lean-auto for translation to the external prover. We consider variants that disable either one:

1. `aesop`: This setting only inputs LEANPREMISE's premises to Aesop as premise applications, omitting calls to Lean-auto or the external prover.

2. `auto`: This setting inputs LEANPREMISE's premises directly to the external prover through Lean-auto without Aesop normalization or premise application.

3. `aesop+auto`: This setting keeps both Aesop and Lean-auto, but does not use premise applications as Aesop rules.

4. `full`: This default setting is the full pipeline described in Section 3.1.

These variants are appealing because they offer cheaper computational cost while still preserving much of LEANHAMMER's ability. Experiments offer insight to the ability of each part of the pipeline (see Section 4.2). We observe that the first three variants may prove theorems that `full` does not, so we also consider `cumul`, which tries all four variants.

We note that other common domain-general automation tactics that take premises as inputs, such as `simp_all`, may be roughly considered a subcase of `aesop` (which we verify in preliminary experiments), so we do not consider them. We also tried using a second-stage model to predict `simp_all` "hints"—whether a premise should be supplied to `simp_all` for preprocessing, and whether it should be applied in reverse direction, but the performance did not increase. We remark that additional automated reasoning tactics in the future may be easily added to our pipeline as a rule of Aesop, similarly to how Lean-auto is added.

## 4 EXPERIMENTS

### 4.1 EXPERIMENTAL SETUP

We extract theorem proofs from Mathlib, and premises from Mathlib, Batteries, and Lean core. In total, we extract 469,965 states from 206,005 theorem proofs, and extract 265,348 (filtered) premises. For each state, there are on average 12.45 relevant premises, giving 5,817,740 (state, premise) pairs in the training set. We randomly hold out 500/500 theorems as valid/test sets, respectively.

We train the model from three base models that were pre-trained for general natural language embedding tasks (Reimers & Gurevych, 2019). These are `small`[5] with 6 layers and hidden size 384 trained from MiniLM-L6, `medium`[6] with 12 and 384 from MiniLM-L12, and `large`[7] with 6 and 768 from DistilRoBERTa-base, respectively. We train our models with learning rate $2e{-}4$, $B = 256$, and $B^- = 3$, found by a hyperparameter sweep. Training the `large` model requires 6.5 A6000-days.

We test LEANHAMMER on proving theorems in (1) our hold-out sets extracted from Mathlib, and (2) the non-Mathlib splits of `miniCTX-v2-test` (Hu et al., 2025). We impose a 10-second time constraint for each call to Zipperposition, and for each theorem a 300-second wall-clock time-out and Lean's default heartbeat limit of 200,000. We tuned the value of $k$ on Mathlib-valid (see Section D.3 of the extended version of this paper (Zhu et al., 2025)), and `full` uses the highest performing combination, which is $k_1 = 16$ premises supplied to Lean-auto (with Aesop priority 10%) and $k_2 = 32$ premises for premise application rules (with Aesop priority 20%). Similarly, we use $k = 16$ for `auto` and `aesop+auto`, and $k = 32$ for `aesop`.

For all experiments and data extraction tasks, we use Lean version v4.16.0. We run a maximum of 16 parallel tests on 16 CPUs with 512GB total memory, so 1 CPU and 32GB are allocated per test theorem. In practice, the actual memory used rarely exceeds 4GB. Each CPU is AMD EPYC 9354 (3.8GHz, 32 cores, 64 threads) or similar.

---

[5]https://huggingface.co/sentence-transformers/all-MiniLM-L6-v2
[6]https://huggingface.co/sentence-transformers/all-MiniLM-L12-v2
[7]https://huggingface.co/sentence-transformers/all-distilroberta-v1

| Premise selector | Model size | Recall (%) | | Proof rate (%) | | | | | |
|---|---|---|---|---|---|---|---|---|---|
| | | @16 | @32 | aesop | auto | aesop+auto | full | cumul | |
| None | — | 0.0 | 0.0 | 16.9 | 9.4 | 16.9 | 16.9 | 16.9 | |
| Random forest* | — | 22.1 | 22.3 | 19.1 | 11.9 | 19.1 | 19.1 | 19.1 | |
| MePo | — | 38.4 | 42.1 | 23.3 | 14.5 | 21.5 | 26.3 | 27.5 | |
| ReProver | 218M | 35.1[†] | 38.7[†] | 11.4 | 12.9 | 20.5 | 12.0 | 22.3 | |
| **LEANPREMISE** (small) | 23M | 59.2 | 67.8 | 23.9 | 19.1 | 25.9 | 27.9 | 31.9 | |
| **LEANPREMISE** (medium) | 33M | 58.6 | 68.1 | 23.1 | 20.1 | 26.1 | 28.5 | 30.7 | |
| **LEANPREMISE** (large) | 82M | **63.5** | **72.7** | **24.1** | **21.3** | **28.5** | **30.1** | **33.3** | |
| **LEANPREMISE** (large) ∪ MePo | | | | 28.9 | 23.9 | 30.3 | 35.9 | 37.6 | |
| **LEANPREMISE** (cumulative) | | | | 27.5 | 25.5 | 31.1 | 34.5 | 37.3 | |
| **LEANPREMISE** (cumulative) ∪ MePo | | | | 30.3 | 27.1 | 32.3 | 38.2 | 39.6 | |
| Ground truth | | | | 27.7 | 33.1 | 37.8 | 41.0 | 43.0 | |

*Performance upper bound, excluding errors. [†]Our definition is slightly different from Yang et al. (2023). See Section C of the extended version of this paper (Zhu et al., 2025).

Table 2: Performance of LEANHAMMER with different premise selectors on Mathlib-test.

| Premise selector | Proof rate using full (%) | | | | | | |
|---|---|---|---|---|---|---|---|
| | Carleson | ConNF | FLT | Foundation | HepLean | Seymour | Average |
| None | 0.0 | 10.0 | 27.3 | 38.0 | 8.0 | 6.0 | 14.9 |
| **LEANPREMISE** (large) | 0.0 | 16.0 | 36.4 | 38.0 | 10.0 | 24.0 | 20.7 |
| Ground truth | 7.1 | 16.0 | 39.4 | 40.0 | 20.0 | 34.0 | 26.1 |

Table 3: Out-of-Mathlib performance of LEANHAMMER on miniCTX-v2-test (Hu et al., 2025) using the large model trained on Mathlib. For other settings than full, see Table 5 of the extended version of this paper (Zhu et al., 2025).

## 4.2 RESULTS

For each theorem, we record the average percentage of ground-truth premises retrieved in the top-$k$ premises (*recall@k*), and the percentage of theorems proven (*proof rate*), shown in Tables 2 and 3. We favor recall over metrics like precision, because a hammer can tolerate irrelevant premises much more than missing important ones.

**LEANHAMMER proves a significant number of theorems.** As shown in Table 2, we find that LEANHAMMER proves a significant proportion of test theorems, with 33.3% proved by the large model in the cumul setting, and 37.3% when accumulated over model sizes. We also test giving ground-truth premises (those that appear in the human-written proof) to LEANHAMMER, which serves as a theoretical best-case scenario of how LEANHAMMER would perform if the models achieved 100% recall, and this proves 43.0% of the theorems. Compared to previous work, LEAN-HAMMER approaches this limit in the settings considered.

**Performance scales with model size and accumulation.** In Table 2 (and Figure 2a of the extended version of this paper (Zhu et al., 2025)), as we increase our model size, for most settings the recall and proof rates also correspondingly increase (e.g., recall@32 increases from 67.8% to 72.7% and full proof rate increases from 27.9% to 30.1%). We also observe that by accumulating across different model sizes or taking the union of neural (our model) and symbolic (MePo) approaches, the proof rate increases much more than scaling the model alone (e.g., full proof rate increases to 34.5% when accumulated), meaning different selectors prove different sets of theorems. More effective methods of ensembling models may be explored in future work.

**LEANHAMMER settings offer different abilities at different costs.** For the settings auto, aesop, aesop+auto, and full, the proof rate roughly increases in this order for all models. This shows that each part of the full pipeline incrementally contributes to the final proof rate. Their mean run times on Mathlib-test are 4.3s, 0.92s, 4.9s, and 6.6s respectively, so the non-full variants are computationally appealing alternatives that recover some of the full performance. We also note

that `cumul` achieves higher proof rates than `full`, so some cases benefit from a partial pipeline (e.g. if the `full` pipeline does not terminate).

**LEANHAMMER shows robust out-of-Mathlib generalization.** As shown in Table 3, the performance on `miniCTX-v2-test` (Hu et al., 2025) is comparable to the performance on Mathlib—the proportion of theorems proven by LEANHAMMER with the `large` selector, out of theorems proven with the ground-truth premises (i.e. the best-case scenario), is 73.5% on Mathlib and 79.4% on `miniCTX` with the `full` pipeline, showing that performance does not decrease (the other settings also have comparable numbers; see Table 5 of the extended version of this paper (Zhu et al., 2025)). We also confirm in the table that if no premises are supplied, the performance is much worse (except for the Foundation split), which indicates that the LEANHAMMER is not just proving trivial theorems.

Across all benchmarks, there are a handful of common patterns characterizing problems that LEANHAMMER fails to solve. Some problems are not solved because LEANPREMISE fails to retrieve necessary lemmas, as can be seen from the fact that the ground truth outperforms all other premise selectors in Tables 2 and 3. Some problems are not solved because they are out of scope for Lean-auto's translation procedure, which can occur when the problem in question contains features from dependent type theory not easily translated to higher-order logic. And some problems are not solved because the solutions require forms of reasoning not supported by Aesop, Zipperposition, or Duper (e.g. induction or arithmetic). Comparatively, it is rare for LEANHAMMER to succeed at proof search with Zipperposition but fail at the proof reconstruction stage with Duper. See Sections D.4 and D.5 of the extended version of this paper (Zhu et al., 2025) for additional analysis.

### 4.3 COMPARISONS

We compare LEANPREMISE against the following existing work: non-LM methods MePo (Meng & Paulson, 2009) and Piotrowski et al. (2023), and LM-based ReProver (Yang et al., 2023). We use a recent adaptation of MePo from Isabelle to Lean (implemented by Kim Morrison), tune its parameters $p$ and $c$ on our evaluation recall@$k$, and apply our premise blacklist. For Piotrowski et al. (2023), we select their random forest model with highest reported performance; in order to overcome errors, we modified its training and evaluation in a way that only gives them unfair advantage, so the reported performance is an upper bound. (See Section C of the extended version of this paper (Zhu et al., 2025) for details of both methods.) We find that LEANPREMISE clearly outperforms either method—for the `large` model, our recall@32 is 73% higher relative to MePo and our `cumul` proof rate is 21% higher (Table 2). Meanwhile, the union of theorems our models and MePo can solve is much higher than each method separately, indicating that symbolic and neural methods have complementary strengths. We believe effective combinations of neural and symbolic methods warrant future investigation.

We retrain ReProver using their training and retrieval scripts, but on our train/valid/test splits and an updated Mathlib version (Zhu et al., 2025, Section C). LEANPREMISE clearly outperforms ReProver (Yang et al., 2023) in terms of recall and proof rate—LEANHAMMER using our `large` model (82M parameters) proves 150% more theorems relative to using ReProver (218M) in the `full` setting and 50% more in the `cumul` setting. We attribute the performance gap to two main factors. First, ReProver focuses on premises used in the next tactic for tactic generation, while LEANPREMISE focuses on finishing the entire proof, so the definitions of ground-truth premises are different (Section 3.2.2). Second, LEANHAMMER uses techniques such as term-style proof extraction, extraction of implicit premises, and better premise signature formatting (Section 3.2). ReProver also uses an $\ell^2$ loss on the cosine similarity for training, rather than our contrastive loss, and we suspect this also contributes to our better performance.

### 4.4 ABLATIONS

Table 4 shows the performance of LEANHAMMER on Mathlib-valid with some components removed: (1) we use a naive data extraction script that (a) uses default pretty-printing options, (b) disables our premise blacklist, and (c) disables collection of premises from `simp` or `rw` calls; (2) we do not sample negative premises during training ($B^- = 0$); and (3) we disable masking positive in-batch premises in the contrastive loss, i.e. the denominator of Equation (2) being simply the sum over all $B(1 + B^-)$ premises in batch. We observe these changes clearly degrade performance.

| Premise selector | Recall (%) | | Proof rate (%) | | | | |
|---|---|---|---|---|---|---|---|
| | @16 | @32 | aesop | auto | aesop+auto | full | cumul |
| LEANPREMISE (medium) | 61.1 | 71.9 | 29.8 | 22.6 | 30.2 | 34.6 | 37.6 |
| + naive data | 57.5 | 66.8 | 29.3 | 20.0 | 28.5 | 33.1 | 35.2 |
| − negatives | 51.8 | 59.5 | 28.6 | 20.0 | 28.4 | 33.0 | 36.8 |
| − loss mask | 59.1 | 69.6 | 29.4 | 21.2 | 29.0 | 34.4 | 38.4 |
| Ground truth | | | 30.8 | 32.0 | 38.4 | 41.2 | 43.6 |
| + naive data | | | 31.2 | 30.0 | 37.0 | 39.8 | 42.4 |

Table 4: Ablation study of LEANHAMMER with different training settings on Mathlib-valid.

Specifically, our data extraction (Section 3.2) is specifically designed with Lean-auto translation in mind, and we observe that settings with Lean-auto have a lower performance with a naive data extraction script. We observe that randomly sampling negative premises and the loss mask (Section 3.3.1) improve performance. (Although the cumul proof rate of LEANHAMMER without the loss mask is higher, we strongly believe this is due to random noise, because all individual settings give lower performances, the recall is lower, and proof rate has higher variance than recall.)

## 5 CONCLUSION

We developed LEANPREMISE, a novel premise selection tool for a hammer in dependent type theory, and combined neural premise selection with symbolic automation to build LEANHAMMER, the first domain-general hammer in Lean. With comprehensive experiments, we show that LEANHAMMER is performant on Mathlib compared to baselines, and generalizes well to miniCTX-v2. LEANPREMISE and LEANHAMMER are designed with accessibility for Lean users in mind, and lay down groundwork for future hammer-based neural theorem proving in Lean.

## ACKNOWLEDGMENTS

We would like to thank Kim Morrison for early discussions about premise selection and implementing a premise selection API and the MePo selector in Lean core, Jasmin Blanchette for proofreading the paper and giving suggestions, Yicheng Qian for discussions about premise selection and building and improving Lean-auto, and Jannis Limperg, Mac Malone, and Joachim Breitner for answering our technical questions. Work partially supported by NSF Grant DMS-2434614 and a gift from Convergent Research.

## REPRODUCIBILITY STATEMENT

We make all code for data extraction, model training, evaluation, and API integration publicly available with an open-source license. Data extraction is open source at `https://github.com/cmu-l3/ntp-toolkit/tree/hammer`, training script is at `https://github.com/hanwenzhu/LeanHammer-training`, premise selection server is at `https://github.com/hanwenzhu/lean-premise-server`, and the Lean premise selection API is partly in Lean core and partly at `https://github.com/hanwenzhu/premise-selection`. We open source both LEANPREMISE and LEANHAMMER as tactics in Lean at `https://github.com/JOSHCLUNE/LeanHammer`. We also release the extracted data and all trained models and baselines.

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
