# OpenReview forum: "Premise Selection for a Lean Hammer"
_ICLR.cc/2026/Conference — ICLR 2026 Oral_

### Official Review · Reviewer_XPxN · 2025-10-23

**Soundness:** 4
**Presentation:** 3
**Contribution:** 3
**Rating:** 8
**Confidence:** 5

**Summary:**

I think this is a well-written and impactful paper that makes a meaningful contribution to the growing intersection between neural methods and formal theorem proving. The authors present LeanHammer, an end-to-end system for automated reasoning in the Lean proof assistant that combines a new neural premise selection model (LeanPremise) with existing components for translation and proof reconstruction. The central innovation lies in designing LeanPremise specifically for dependent type theory, and in showing that it can dynamically adapt to user-specific contexts including both large existing libraries and locally defined lemmas. This is a significant advance over prior premise selection systems for Lean, which were often static or trained on limited corpora.

Empirically, the paper is strong. The authors demonstrate that LeanPremise enables LeanHammer to solve 21% more goals compared to existing premise selectors, and that it generalizes effectively across diverse mathematical domains. I found the experiments to be thorough and convincing, and the evaluation metrics appropriate for the problem setting.

**Strengths:**

Strengths:

Clear Motivation and Novelty: The paper is well-motivated in the broader context of neuro-symbolic reasoning. The idea of combining neural retrieval (premise selection) with symbolic reasoning (hammer backends) is well-established in Isabelle and Coq, but its extension to dependent type theory in Lean is non-trivial and timely.

Practical Integration: I appreciate that the authors actually built an integrated hammer for Lean! Something the community has wanted for years. The engineering effort here is nontrivial and valuable.

Adaptive Context Handling: The dynamic adaptation to local user contexts is, in my view, one of the paper’s most important contributions. It shows a genuine understanding of how theorem proving workflows operate in practice.

Strong Empirical Results: The 21% improvement in goal-solving rate is not just statistically significant — it’s practically meaningful. The ablation and generalization studies make a solid case for the effectiveness of LeanPremise.

Clarity and Presentation: The writing is clear, structured, and accessible even to readers not deeply familiar with Lean. The paper situates itself well within the literature on neural premise selection and hammer systems.

**Weaknesses:**

Some minor weaknesses that i thought

1. Model Details and Reproducibility: While the authors provide a solid description of the LEANPREMISE training pipeline, the architecture details of the encoder model (layer sizes, tokenization, embedding pooling, etc.) are somewhat brief. The training section mentions that they fine-tune MiniLM and DistilRoBERTa models, but additional detail on input formatting or tokenizer alignment with Lean syntax would help others reproduce or extend this work.

2. Limited Discussion of Failure Cases: The experiments are comprehensive, but it would strengthen the paper to include more qualitative analysis of where the hammer fails e.g., typical patterns of unsuccessful proofs, sensitivity to theorem complexity, or when external provers (like Zipperposition) fail to reconstruct. Understanding these limits could guide future improvements to both premise selection and proof reconstruction.

**Questions:**

1. I really liked that LeanPremise can incorporate locally defined facts at runtime. Could the authors clarify how new lemmas are embedded on-the-fly? For example, are these signatures tokenized and passed through the encoder server dynamically, or is there a cached retriever that updates incrementally?

2. Did the authors explore hybrid ranking (e.g., mixing similarity-based and logic-based retrieval) or reinforcement-based refinement of premise scores?

3. The paper describes four hammer variants (aesop, auto, aesop+auto, full). In practice, how sensitive is LeanHammer’s success rate to the choice of variant and to timeout parameters (e.g., 10 s for Zipperposition)? Would adaptive scheduling of these variants yield further gains?

---

> ### Author Response · Authors · 2025-11-20
>
> **W1**: Regarding the architecture details of the encoder model, we just use pretrained models (small: https://huggingface.co/sentence-transformers/all-MiniLM-L6-v2, medium: https://huggingface.co/sentence-transformers/all-MiniLM-L12-v2, large: https://huggingface.co/sentence-transformers/all-distilroberta-v1). There is no Lean-specific tokenizer alignment because we use pretrained models. We are happy to clarify these details in the paper itself.
>
> **W2**: As mentioned in the overall response, we do discuss some details related to failure analysis in our appendices, but based on this feedback, we plan to move some of the discussion of the failure analysis to the main text in the next couple of days. We also plan to introduce some more qualitative information (namely, that Zipperposition, Duper, and Aesop share the weakness of being unable to perform induction and arithmetic) to help guide future improvements.
>
> **Q1**: Signatures are dynamically computed on the user side, then sent to our hosted API server for embedding. The server pre-caches all Mathlib and Lean core premises at fixed release versions, so the user side does not need to upload them. User-defined premises have embeddings stored in a cache so every formalization project only needs to be embedded once by some user.
>
> **Q2**: We have not yet explored either of these avenues in depth, though we believe both would be valuable directions for future research. The results in Table 2 showing that LeanPremise (large) U MePo outperforms both LeanPremise (large) and MePo on all settings suggest that there are concrete gains that might be obtained by hybrid ranking. Using the verifier feedback to improve the premise selector (e.g., via reinforcement learning) is certainly an interesting future direction to explore.
>
> **Q3**: Tables 2 and 5 contain information on the impact of LeanHammer’s success rate to the choice of variant, and Figure 2B in Appendix D.1 gives runtime information. To summarize, the full variant broadly performs better than LeanHammer’s other variants, but the fact that cumul outperforms full indicates that some cases benefit from a partial pipeline. Regarding timeout parameters, the fact that most problems solved by LeanHammer are solved in under 5 seconds indicates that increasing Zipperposition’s 10 second timeout is unlikely to significantly improve LeanHammer’s performance.
>
> We have not yet explored adaptive scheduling for these variants, though the fact that some cases benefit from a partial pipeline does indicate that adaptive scheduling may yield improvements. On the other hand, it may turn out that creating a portfolio of sufficiently different variants and running them in parallel works just as well. We feel that both approaches merit further exploration.

---

> > ### Comment · Reviewer_XPxN · 2025-11-28
> > **Response**
> >
> > Great, thanks for answering my questions! i keep my score however: )!

---

### Official Review · Reviewer_rKRM · 2025-10-29

**Soundness:** 3
**Presentation:** 3
**Contribution:** 3
**Rating:** 8
**Confidence:** 3

**Summary:**

The paper introduces LEANPREMISE, a learned premise retriever for Lean, and integrates it into LEANHAMMER, an automated “hammer” tactic for Lean. The system retrieves relevant lemmas, translates to higher-order logic, calls ATPs, and reconstructs proofs back in Lean. The authors claim this is the first practical, general-purpose Lean hammer that normal Lean users can call. They train LEANPREMISE on hammer-style data (including implicit premises and both tactic/term proofs), and show strong gains on Mathlib and an out-of-distribution benchmark (miniCTX-v2), with substantially higher recall and proof rates than baselines like MePo or ReProver.

**Strengths:**

1. The pipeline design is careful and realistic for Lean’s dependent type theory. The data extraction method (capturing all premises truly used in a proof state, not just the next tactic) is well motivated, and ablations support it. The system also supports dynamically including a user’s local lemmas at inference time via FAISS indexing.

2. LEANPREMISE achieves much higher recall@32 (≈70%+) than MePo and ReProver, and when plugged into LEANHAMMER, proof rates jump from ~20–27% (baselines) toward ~33%+ for Mathlib-test, and near 40% with ensembles. It also transfers to miniCTX-v2, suggesting generalization beyond Mathlib.

3. A turnkey hammer for Lean is genuinely impactful for interactive theorem proving and AI-for-math. The work is clearly written, includes ablations, and promises to release code, data, and models.

**Weaknesses:**

1. The paper does not compare against a simple LLM as premise suggester baseline: prompt a modern LLM with the Lean goal and ask it to list ~32 relevant lemmas from Mathlib, then feed those to the same hammer. Because the hammer mainly needs high recall and tolerates many irrelevant premises, this is an important baseline. It would clarify how much LEANPREMISE beats “just ask an LLM.”

2. Some miniCTX-v2 domains are still barely solved, and it would help to briefly analyze where the pipeline fails (missing premises vs. reconstruction failures).

**Questions:**

How robust is LEANHAMMER to hallucinated lemmas (names that don’t exist)? Do they just get ignored like harmless false positives?

What exact hardware produced the reported latencies, and how does latency scale with many user-defined lemmas?

For hard miniCTX-v2 splits, what is the main blocker: retrieval, ATP search, or Lean reconstruction?

---

> ### Author Response · Authors · 2025-11-20
>
> **W1**: We appreciate that gathering this data could help quantify the importance of using targeted training methods over "just asking an LLM.” Though we haven’t performed the experiments necessary to quantify how much worse just asking an LLM would be, we expect it would be significantly worse. We prompted ChatGPT 5.1 to provide a list of 32 premises for the example in Appendix B (making sure to clarify that the example pertains to Lean version v4.16.0). It returned the following list:
>
> [associated_iff_dvd_dvd*, associated.symm*, associated.trans*, associated_mul_mul*, associated_of_dvd_dvd, associated.eqv*, associated.refl*, dvd_gcd_right*, gcd_dvd_left, gcd_dvd_right, dvd_gcd_iff, gcd_rec*, gcd_comm, gcd_assoc, gcd_mul_left, gcd_mul_right, dvd_mul_of_dvd_left, dvd_mul_of_dvd_right, dvd_trans, dvd_refl, dvd_antisymm, dvd_of_mul_dvd_mul_right*, dvd_of_mul_dvd_mul_left*, normalize_gcd, normalize_associated, associated_normalize, associated_of_mul_associated_mul_right*, associated_of_mul_associated_mul_left*, gcd_eq_left_iff_dvd*, gcd_eq_right_iff_dvd*, gcd_dvd_iff*, dvd_gcd_of_dvd_of_dvd*].
>
> Of these, the 16 names marked with asterisks were hallucinated. Filtering these premises and supplying the remainder to LeanHammer was insufficient to allow LeanHammer to solve the example.
>
> **W2**: As mentioned in the overall response, we do discuss some details related to failure analysis in our appendices, but based on this feedback, we plan to move some of the discussion of the failure analysis to the main text in the next couple of days.
>
> **Q1**: Our pipeline doesn’t produce any hallucinated names, but constants or theorems that can’t be processed by LeanHammer for whatever reason (e.g. because they are outside the scope of Lean-auto’s translation procedure) are ignored.
>
> **Q2**: Regarding hardware, we run a maximum of 16 parallel tests on 16 CPUs with 512GB total memory. So 1CPU and 32GB are allocated per test theorem. In practice, the actual memory used rarely exceeds 4GB. Each CPU is AMD EPYC 9354 (3.8GHz, 32 cores, 64 threads) or similar.
>
> Regarding latency, for many user-defined lemmas, indeed the initial latency scales linearly, because the signature for each user-defined lemma needs to be calculated and embedded. However, this is only a one-time cost because the signature is cached on the user side, and the embedding is cached on the server. Moreover, all Mathlib and Lean core premises are pre-embedded on the server. For a user tool we currently recommend capping the number of user-defined lemmas to 2048.
>
> **Q3**: For the Carleson split, we marked LeanHammer’s proofrate as 0% because the pipeline resulted in an unexpected error (Appendix D.2). For other splits, we broadly saw that proof reconstruction in Lean was not a significant blocker (in Appendix D.5, we note that under 2% of Mathlib-test theorems can be proven by Zipperposition but not Duper). Beyond that, in specific cases where ATP search fails, we note that it is difficult to tell whether it is failing because it wasn’t given a necessary premise or because it fundamentally lacks the ability to solve the underlying problem (e.g. because the underlying problem requires induction or arithmetic).
>
> For a more detailed look at the miniCTX-v2 splits, we recommend looking at Table 5 in Appendix D.2. This table shows LeanHammer’s performance on each miniCTX-v2 split both with LeanPremise and with the ground truth. On splits where LeanHammer’s performance with LeanPremise is close to LeanHammer’s performance with the ground truth, we might infer that ATP search is the bigger blocker, and on splits where LeanHammer’s performance with LeanPremise is substantively worse than LeanHammer’s performance with the ground truth, we might infer that retrieval is the bigger blocker.

---

> > ### Comment · Reviewer_rKRM · 2025-11-25
> > **No more questions**
> >
> > I would like to thank authors for their responses. I've no more questions.

---

### Official Review · Reviewer_u2zP · 2025-10-29

**Soundness:** 3
**Presentation:** 2
**Contribution:** 2
**Rating:** 4
**Confidence:** 3

**Summary:**

This paper introduces a novel premise selector, LeanPremise, tailored for premise selection for hammer use.
The authors fine-tune a textual encoder on carefully extracted (state, premise) data pairs with an InfoNCE-like loss, and rank premises based on embedding similarity.
By integrating LeanPremise with Aesop, Lean-auto and Duper, they also build a LeanHammer, which improves the accumulate proof rate from 27.5% (baseline, using MePo as the premise selector) to 33.3% on Mathlib-test. Notably, LeanHammer is currently accessible by simply using the corresponding tactic in Lean.

**Strengths:**

Significance: This work fills the blank that there is no hammer tool (premise selection + translate to external language for ATP/SMT solver application + translate back) in Lean. The proposed LeanHammer tool is practical. It has been built as a tactic in Lean, and I personally have been using it for daily hands-on proofs.

Clarity: The pipelines of LeanHammer and LeanPremise are clearly stated. Details such as how to handle the signature are enough covered.

Quality: The performance of LeanPremise is verified extensively on multiple datasets and various experimental settings, compared with many previous methods as baselines.

Originality: (1)The data extraction pipeline is tailored for hammer use in many details, such as using all premises for the current goal rather than only for the next tactic. (2)  The use of InfoNCE loss for training the LeanPremise encoder, instead of binary-class MSE loss used by ReProver, is quite reasonable. (3) The LeanHammer pipeline not only makes use of current strong tools such as Aesop and external theorem provers like Zipperposition, and also alllows for adding additional automation tactics in the future as a rule of Aesop.

**Weaknesses:**

1. The conceptual novelty of this work is limited. The overall design of LeanHammer mainly follows that of the Sledgehammer in Isabelle, and the LeanPremise is a standard embedding similarity based retrieval method, similar to what ReProver does. The contribution is mainly engineering rather than methodological.
2. The writing of the paper should be further checked. There are typos and confusing statements that could harm the understanding of the paper. For example, on page 6, $p_{i}^{+}$ should be changed to $p_{i}^{-}$ in the sum for negative premises in the denominator in the loss function expression, and the definition $$\mathcal{N}_{i}=\{p_{i}^{+}\}_{i}\cup\{p_{ij}^{-}\}_{ij}\setminus\mathcal{P}_{s_{i}}^{+}$$ should be simplified to $\{p_{ij}^-\}_{j=1}^{B^-}$ directly.
3. The final performance achieved by LeanHammer is not that impressible compared to the performance of today's LLM theorem provers. On miniCTX-v2, the current SOTA Seed-Prover(light mode) can prove 81.8% of the theorems without accessing any external ATP/SMT solvers. In contrast, LeanHammer could only achieve 26.1% even with ground truth premises, let alone the 20.7% using LeanPremise.

**Questions:**

1. I am most curious about whether implementing such a hammer to make use of external ATP/SMT solvers can help with LLM theorem provers. I will appreciate if the authors can provide some examples where the current open-weights SOTA models such as Goedel-Prover-V2-32B fail in pass@k(k=32 or so), but LeanHammer can help the model find a proof.

2. Have you tried to add automation tactics other than simp_all, such as linarith, omega, native_decide, etc. into the pipeline? My experience is that these tactics can usually help solve goals that Aesop, simp_all and external ATPs cannot solve. I agree with that simp_all can not help improve further from Aesop.

---

> ### Author Response · Authors · 2025-11-20
>
> **W1**: LeanHammer does share design features with Isabelle’s Sledgehammer insofar as both are hammers, and therefore both contain premise selection, translation to external provers, and proof reconstruction. However, the use of LeanPremise’s recommended premises as unsafe Aesop rules more closely resembles MagnusHammer’s approach of bypassing translation to external provers, and LeanHammer’s integration of the goal translation and proof reconstruction pipeline as part of Aesop’s backtracking tree search is, to our knowledge, completely novel.
>
> Likewise, LeanPremise does bear similarities to ReProver insofar as both are neural retrieval methods, but we found that their differing goals (premises for next-tactic generation vs. premise selection explicitly targeting hammer use) lead to different data extraction methodologies and significantly different results. Please refer to the paper for a precise comparison of LeanPremise and ReProver, which we discuss in several places. Since other premise selection tools either don’t target hammer use (e.g. ReProver), use traditional machine learning techniques (e.g. CoqHammer), or don’t target dependent type theory (e.g. MagnusHammer), we believe that documenting the training and data extraction techniques which optimize LeanPremise for LeanHammer’s use has value beyond the engineering work required to implement said techniques.
>
> **W2**: We thank the reviewer for carefully checking our notation. The denominator of Eq. (2) indeed contains the typo and has been updated in the paper. However, the set of negative premises $\mathcal{N}\_{i}$ used in the loss for each state $s\_i$ is actually larger than the set of sampled negative premises $p\_{ij}^-$ for state $s\_i$, because $\mathcal{N}\_{i}$ also includes premises from other indices $i’$ in the batch. In fact, every state $s\_i$ is associated with all $B(1+B^-)$ premises in the entire batch minus the positive ones $\mathcal{P}\_{s\_i}^+$. This is a standard practice (in-batch negative sampling) and is used in previous work like ReProver and Magnushammer.
>
> **W3**: We believe that LeanHammer and whole-proof reasoning models such as Seed-Prover ultimately address different problems. LeanHammer is intended to be a user-facing tool which is broadly accessible and fast enough to be used interactively. All of LeanHammer’s settings have mean run times well under 10 seconds, whereas Seed-Prover’s light mode is said to complete in 1-2 hours. Furthermore, the approaches are complementary: Seed-Prover could call LeanHammer since LeanHammer is exposed as a tactic.
>
> **Q1**: We haven’t explored the interaction between LeanHammer and LLM theorem provers, but prior work in Isabelle suggests that LLM-based provers and hammers are highly complementary (https://dl.acm.org/doi/10.5555/3600270.3600878). LLM-based provers without retrieval are powerful but limited by their context, while hammers are relatively straightforward but are designed specifically to enjoy the benefits of having a comprehensive view of the dependent libraries. Based on this, we are optimistic that exploring the interaction of LeanHammer and LLM theorem provers will be fruitful future work.
>
> **Q2**: We fully agree that there are other tactics complementary to Aesop and Duper in that they are well-equipped to solve different sorts of problems. We are actually currently in the process of integrating `grind` (a powerful tactic that should approximately subsume `linarith` and `omega` when used correctly) into LeanHammer, and we are very optimistic that the integration will yield significant improvements, particularly on problems involving arithmetic.

---

### Official Review · Reviewer_cvrg · 2025-11-01

**Soundness:** 3
**Presentation:** 3
**Contribution:** 2
**Rating:** 6
**Confidence:** 4

**Summary:**

The paper introduces LEANPREMISE, an LM-based premise selector tailored for use with a hammer in Lean’s dependent type theory, and integrates it with Aesop, Lean-auto (DTT to HOL translation to external ATPs), and Duper to build LEANHAMMER, "the first end-to-end domain-general hammer for Lean." The selector standardizes premise/state serialization, trains with a masked contrastive objective, and dynamically incorporates user/local premises. On Mathlib and miniCTX-v2, the system improves proof rates over prior premise selectors and shows complementary behavior with MePo. The authors provide ablations and claim open-sourced code, data, and trained models.

**Strengths:**

1. Positions premise selection explicitly for a Lean hammer, not just for next-tactic generation, with hammer-aware data extraction (term + tactic proofs; collecting implicit simp/rw premises; signature normalization that avoids notation brittleness). This is a concrete, useful framing that, as far as I know, prior Lean premise selectors did not target.

2. Dynamic augmentation with project-local premises and server-side caching for low-latency retrieval is practically novel for Lean users.

3. Clear methodology for training: masked InfoNCE with explicit negatives; ablation shows each training choice (negatives, loss mask, data extraction) matters.

4. If robust, first domain-general hammer for Lean is a meaningful community contribution, potentially changing day-to-day Lean workflows and enabling downstream NTP systems to "call a hammer" in Lean.

**Weaknesses:**

1. The paper retrains/ports baselines (e.g., MePo to Lean; ReProver on the authors’ splits) and changes data extraction relative to next-tactic systems; while understandable, this complicates strict novelty/efficacy attribution. It would be good to expand baseline re-implementation details and release scripts to reproduce exact comparison settings (hyperparameters, filters, blacklists) and the exact evaluation splits.

2. Success depends on Lean-auto translation quality and Duper reconstruction once Zipperposition returns premises. It would be good to have more failure analyses, for example to provide breakdowns of (a) external ATP success vs. Duper reconstruction failure, (b) cases where Aesop premise applications alone succeed vs. Lean-auto is essential, and (c) how often missing implicit vs. explicit premises is the root cause.

3. Filtering 479 "basic logic" theorems and aggressive fully-qualified, notation-free signatures are sensible, but please quantify how each filter/format affects recall, ATP success, and end-to-end proofs on a held-out set (beyond the aggregate ablation), and whether performance is brittle to Mathlib version changes.

4. The reported metrics could be richer. For example, consider reporting proof length, number of premises used, and time-to-first-proof distributions, to understand proof mechanics, as well as the effect of k (and of per-step dynamic k) across problem difficulty buckets, etc.

**Questions:**

1. While “domain-general” within Lean, it’s unclear how easily the approach ports to other proof assistants like Rocq or Isabelle. A cross-assistant pilot (even small-scale or case-by-case) would strengthen the generality claim and offer insights for future work to extend the approach.

2. In the “ground-truth premises” oracle runs, how many of those premises are actually required for the found proof vs. incidental? Could a minimal-premise oracle raise the theoretical ceiling?

3. When users work in private libraries with heavy custom notations (even if you print fully-qualified constants), how often do premise embeddings drift enough to harm retrieval? Any quantitative OOD tests beyond miniCTX-v2?

---

> ### Author Response · Authors · 2025-11-20
>
> **W1**: We are happy to release all scripts necessary so that readers can reproduce our experiments. MePo has already been added to Lean core (see https://github.com/leanprover/lean4/blob/master/src/Lean/LibrarySuggestions/MePo.lean), so this is already publicly available. For ReProver, we use the same hyperparameters, data extraction script, and training script as the official ReProver repo (see https://github.com/lean-dojo/ReProver/tree/main/retrieval). The only difference is we made sure to run their data extraction script on the same version of Lean and Mathlib as ours (version v4.16.0) and use the same train/valid/test splits. As mentioned in the overall response, we are committed to making all aspects of LeanHammer and LeanPremise open source, including details such as hyperparameters and blacklists, which should make replication possible for interested readers.
>
> **W2**: Regarding (a), external ATP success vs. Duper reconstruction failure, we distinguish between these in Appendix D.5.
>
> Regarding (b), the set of problems that LeanHammer solves in the Aesop setting is the set of problems where Aesop premise applications alone succeed. The difference between LeanHammer’s performance in the Aesop setting and the full setting comes from the set of problems where Lean-auto is essential.
>
> Regarding (c), we acknowledge that the data we provide does not directly indicate how often missing premises are the proximate cause of failure. Recall gives information on LeanPremise’s efficacy in identifying necessary premises, but in cases where the external ATP fails, it is difficult to tell whether it is failing because it wasn’t given a necessary premise, or whether it is failing because it fundamentally lacks the ability to solve the underlying problem (e.g. because the underlying problem requires induction or arithmetic).
>
> **W3**: We agree that the analysis of our data extraction procedure may be improved by more fine-grained experiments showing the independent impact of each difference between our hammer-aware data extraction procedure and the naive data extraction procedure we compare against in Table 4. We would be happy to add this analysis to the camera-ready version.
>
> We have not directly measured the impact of Mathlib version changes on LeanHammer’s and LeanPremise’s performance. However, we have kept LeanHammer and LeanPremise up to date on the latest stable version of Lean and Mathlib, so we can comment on our anecdotal experience. Generally, we have found that it is possible to reuse models corresponding to older versions of Mathlib without an immediately obvious downgrade in performance, but that it is very important to keep cached embeddings of Mathlib theorems up to date. New theorems are added to Mathlib frequently, and it is not uncommon for old theorems to be renamed or deprecated. The cost of calculating signatures and embedding all theorems changed between Mathlib versions is significant enough to be noticeable. As a consequence, our tool caps the number of user-defined theorems (including theorems from a version of Mathlib that the server hasn’t pre-embedded) at 2048.
>
> **W4**: In Appendix D.1, we do report runtime distributions, but we don’t measure proof length or number of premises used in the proofs output by LeanHammer. We believe that the metrics we measure and report on (recall, proof rate, and runtime) characterize the most important aspects of LeanPremise and LeanHammer’s practical usability, but we agree that more metrics could provide additional insight.
>
> **Q1**: We agree with the reviewer that expanding LeanHammer to other proof assistants can further validate the approach. We consider this future endeavour a highly fruitful one, and made significantly easier by our Lean-specific tool development contributions.
>
> **Q2**: Generally, we expect that the premises used in an actual proof will all be needed in a proof found by Zipperposition or Duper, but the reviewer is right to note that some of the facts can be supplied automatically by tools like `simp` which bring in their own premises. It would be interesting to explore in future research how variations on our premise selection may interact with automation that can supply some of its own facts.
>
> **Q3**: All notations, including custom user notations, are disabled during printing.

---

> > ### Comment · Reviewer_cvrg · 2025-11-25
> >
> > Thanks for the detailed responses! All my concerns have been addressed and all questions are answered. I would like to further raise my score to 8 to show stronger support, with the Contribution subscore raised to 3. Thanks for the good work!

---

### Author Response · Authors · 2025-11-20

We are grateful to the referees for their thoughtful reviews. We are pleased that on the whole, the reviewers found our work to be well-motivated and practical. Several reviewers raised questions about details of our methodology (e.g. hardware used, hyperparameters, blacklists, etc.) and expressed a desire to see more analysis of where and how the hammer fails.

Concerning the methodology, we are committed to making all aspects of LeanHammer and LeanPremise open source and as transparent as possible. We are optimistic that, should the paper be accepted, many questions can be simultaneously addressed by including links to:
- Our data extraction scripts
- Our model training scripts
- The code that implements the LeanPremise server
- The implementation of LeanHammer itself
- Our exact evaluation splits

In addition to the above, we will modify the paper to address and clarify particular points mentioned by the reviewers (e.g. we will fix the typo discovered by u2zP, we will include hardware details requested by rKRM, and we will expand on the architecture details of the encoder model as requested by XPxN).

Concerning the failure analysis, we agree that some additional comments in the main text would be beneficial. We do discuss some details related to failure analysis in our appendices. Appendix D.2 comments on the Carleson split, Appendix D.4 provides some statistics on the lengths of proofs that LeanHammer is (and isn’t) able to replace, and Appendix D.5 gives statistics on which components tend to be responsible for LeanHammer failures. But we agree that there should be more discussion of this in the main text, so in the next couple of days, we will modify the paper accordingly.

Update: A revised version of the paper containing the aforementioned changes has been uploaded.

---

> ### Author Response · Authors · 2025-11-29
>
> We affirm that we do not know who our reviewers are and had no interaction with them outside the system. We want to note for the record that reviewer u2zP changed their score to 6 within a few hours and cvrg increased their score to 8 in response to our rebuttal. Finally, we want to thank the area chairs for making the best of a difficult situation.

---

### Meta-Review · Area_Chair_Z4AD · 2026-01-07

**Summary:**

This paper introduces a neural premise selection system, LeanPremise, and then combines it with existing translation and proof reconstruction components to create an end-to-end, domain-general LeanHammer. The paper also provides a user-facing tactic interface that dynamically processes new premises. Comparison with Random Forest, MePo, and Reprover indicates that the proposed method outperforms in both premise selection recall and proof rate.

**Reviewer Concerns:**

Most of the reviewer concerns are addressed, including reproducibility, failure analysis, and confusing statements in the manuscript from all reviewers.

**Reviewer Scores:**

Reviewer cvrg has clearly indicated an increase in score from 6 to 8. Reviewer u22P is also likely to raise the score, as her/his concerns have largely been addressed. The other two reviewers have already given high scores of 8.

---

### Decision · Program_Chairs · 2026-01-26

Accept (Oral)